# The level of adherence to best-practice guidelines by interprofessional teams with and without acute care nurse practitioners in cardiac surgery: A study protocol

**Li-Anne Audet**[1]*, **Mélanie Lavoie-Tremblay**[2], **Éric Tchouaket**[3], **Kelley Kilpatrick**[1,4,5]

**1** Ingram School of Nursing, Faculty of Medicine and Health Sciences, McGill University, Montreal, QC, Canada, **2** Faculté des Sciences Infirmières, Pavillon Marguerite-d'Youville, Université de Montréal, Montreal, QC, Canada, **3** Département des Sciences Infirmières, Université du Québec en Outaouais, Saint-Jérôme, QC, Canada, **4** Centre Intégré Universitaire de Santé et de Services Sociaux de l'Est-de-l'Île-de-Montréal-Hôpital Maisonneuve-Rosemont (CIUSSS-EMTL-HMR), Montreal, QC, Canada, **5** Susan E. French Chair in Nursing Research and Innovative Practice, Ingram School of Nursing, Faculty of Medicine and Health Sciences, McGill University, Montreal, QC, Canada

* li-anne.audet@mail.mcgill.ca

**Data Availability Statement:** No datasets were generated or analysed during the current study. All

## Abstract

### Background

Acute care nurse practitioners (ACNPs) in postoperative cardiac surgery settings provide significant benefits to patients and organizations. Recent studies have suggested that ACNPs increase the level of adherence to best-practice guidelines by interprofessional teams. It is however, unknown whether interprofessional teams with ACNP are associated with higher levels of adherence to best-practice guidelines compared to interprofessional teams without ACNPs. Furthermore, no extraction tool is available to measure the level of adherence to best-practice guidelines by interprofessional teams in postoperative cardiac surgery settings. This project aims to measure and examine the level of adherence to best-practice guidelines of interprofessional teams with and without ACNPs in a postoperative cardiac surgery setting in Québec, Canada.

### Methods

A retrospective observational study will be conducted of 300 patients hospitalized between January 1, 2019 and January 31, 2020 in a postoperative cardiac surgery unit in Québec, Canada. Data will be collected from patient health records and electronic databases. An extraction tool will be developed based on systematic review of the literature, and will include best-practice guidelines and confounding variables related to patient and interprofessional teams' characteristics. Content and criterion validation, and a pilot-test will be conducted for the development of the tool. A multivariate linear regression model will be developed and adjusted for confounding variables to examine the association between interprofessional teams with and without ACNPs, and level of adherence to best-practice guidelines by those teams.

relevant data from this study will be made available upon study completion.

**Funding:** This project is funded by the Réseau de recherche en intervention en sciences infirmières du Québec (RRISIQ). LAA holds doctoral scholarships from the Fonds de recherche du Québec – santé as well as from the Québec Ministry of Education (Ministère de l'Éducation et de l'Enseignement Supérieur du Québec). ET holds a career award from the Fonds de recherche du Québec – Santé (Junior 2 Research Salary Award). KK holds a career award from the Fonds de recherche du Québec – Santé (Senior Research Salary Award) and holder of the Susan E. French Chair in Nursing Research and Innovative Practice. No funding sources were involved in the study conception and design, data collection, analysis, interpretation, or in the final decision to submit this manuscript for publication.

**Competing interests:** The authors have declared that non competing interests exist.

## Discussion

This project represents the first study to measure and examine the level of adherence to best-practice guidelines by interprofessional teams with and without ACNPs in a postoperative cardiac surgery setting. The findings of this project will generate empirical data focusing on the contribution of ACNPs within interprofessional teams, and ultimately enhance the delivery of high quality and evidence-based care for patients and families.

## Background

Cardiac surgeries are one of the leading types of surgical procedures performed internationally. In 2017, approximately one million patients underwent cardiac surgery, the majority of whom reside in developed nations [1]. In 2018, the Society of Thoracic Surgeons (STS) reported 287,872 cardiac surgeries and procedures performed in the United States [2]. In Canada, 42,989 coronary artery bypass grafts (CABG) and 7,186 valve repairs were performed between 2013 and 2016 [3, 4]. In 2014, Germany had the highest rate of coronary revascularization procedures among the European union with 453 procedures per 100,000 population, followed by Austria (324 procedures per 100,000 population), Croatia (324 procedures per 100,000 population), and Lithuania (319 procedures per 100,000 population) [1].

In the postoperative phase after cardiac surgery, many patients are at high risk of developing adverse events and postoperative complications such as myocardial infarction and wound infection [5]. These complications are associated with higher risk of mortality and the development of comorbidities (e.g., heart failure), as well as a decrease in well-being and quality of life of patients and families [6]. For healthcare organizations, adverse events and postoperative complications are associated with a longer length of stay at the hospital, higher rates of readmission in intensive care units, higher rates of surgical re-exploration, and increased costs [7, 8].

To prevent the risk of adverse events and postoperative complications for patients and families, healthcare organizations and research teams internationally have developed best-practice guidelines [7, 9]. Best-practice guidelines are defined as evidence-based practice aimed to ensure the performance of interprofessional teams, to enhance the quality and safety of the care given to patients and families. In postoperative cardiac surgery settings, best-practice guidelines related to pharmacotherapy, laboratory tests, clinical indicators, and lifestyle promotion (e.g., diet, cardiac rehabilitation) have been developed and implemented in healthcare organizations [2].

A consensus in the literature supports the importance of a high level of adherence by interprofessional teams to best-practice guidelines to ensure their efficacy [7, 10]. Adherence to best-practice guidelines is defined as the achievement of the guideline, as well as the associated interventions performed by clinicians [11]. In surgical settings (e.g., neurosurgery, orthopedic surgery) international research teams [12–15] have developed composite scores to examine the level of adherence to best-practice guidelines by interprofessional teams. Their findings suggest that a higher level of adherence is associated with a higher quality of care given to patients and families.

In postoperative cardiac surgery settings, recent studies [5, 16] identified a significant association between a high level of adherence to best-practice guidelines by interprofessional teams and a lower risk of postoperative complications and adverse events for patients. Other studies [17] have shown a significant association between a lower level of adherence to best-

practice guidelines by interprofessional teams and a higher risk of postoperative complications and adverse events for patients. Larrazet et al. (2014) conducted a longitudinal study of 144 patients who died after cardiac surgery. The authors identified a significant association between a lack of adherence to best-practice guidelines by interprofessional teams, and a higher risk of mortality, highlighting the importance of the high adherence to best-practice guidelines by interprofessional teams [17].

Internationally, there is variability in the scope of practice and level of autonomy of acute care nurse practitioners (ACNP) [18, 19]. Recent studies have examined the practice of nurse practitioners (NP) within primary and acute care settings and suggest that their practice could increase the level of adherence to best-practice guidelines by interprofessional teams [18, 19]. NP's evidence-based practice is believed to support care providers' adherence to best-practice guidelines within the areas of pharmacotherapy, clinical indicators, and lifestyle promotion. NPs also enhance collaboration and communication among teams' members and facilitate continuity of care, contributing to the higher adherence to best-practice guidelines by interprofessional teams [18, 20].

Based on the International Council of Nurses guidelines, the NP is an advanced practice nursing role, based on a graduate or postgraduate education and an in-depth nursing and clinical expertise (e.g., prescription of the pharmacotherapy, diagnosis, etc.) [21–23]. In postoperative cardiac surgery, ACNP have been implemented in interprofessional teams and performed activities such as the clinical and psychosocial assessment of patients, the prescription and management of pharmacotherapy, laboratory tests and clinical interventions, lifestyle promotion, teaching of coping strategies, preparation for hospital discharge, and the management of consultations and external resources [24]. International research teams [23, 25] have examined the practice of ACNP within postoperative cardiac surgery settings. They identified significant associations between the ACNP' practice and higher patient satisfaction during their hospital stay, higher rates of patients participating in cardiac rehabilitation programs, and reduced length of stay at the hospital after the cardiac surgery. In contrast, other research teams [21, 26] did not identify significant associations between ACNP practice and patient and organizational outcomes within postoperative cardiac surgery settings. More specifically, the authors suggested that the difficulty of recruiting participants, the high attrition rate, the small sample sizes, and the inability to adjust the data analysis for confounding variables related to the patient and interprofessional teams' characteristics were important limits contributing to the non-significant findings [21, 26].

Systematic reviews of randomized controlled trials (RCT) [21, 23] have been conducted to understand the inconsistent findings in the current literature. These systematic reviews have identified three important limits contributing to the inconsistent findings of the current literature. Firstly, although several RCTs have focussed on the efficacy of ACNPs for patients and healthcare organizations, less attention has been paid to the efficacy of these providers for interprofessional teams. The qualitative study by Reich et al. (2018) suggested that the implementation of ACNPs within interprofessional teams increases the level of adherence of those teams to best-practice guidelines in postoperative cardiac surgery settings, and represents an underlying factor contributing to the efficacy of these teams [19]. However, these propositions have not been subjected to statistical validation, thus it is unknown if interprofessional teams with ACNP are associated with higher levels of adherence to best-practice guidelines, compared to interprofessional teams without ACNP. Additionally, to date no validated extraction tool to measure the level of adherence to best-practice guidelines of interprofessional teams with and without ACNPs in postoperative cardiac surgery has been located.

Secondly, systematic reviews of randomised controlled trials (RCTs) have revealed that many variables (e.g., severity of illness) influence the effect of ACNP practice on patient and

organizational outcomes in postoperative cardiac surgery settings. These variables have not been adequately controlled in current RCTs [21, 23]. Methodological limits of the existing RCTs, such as the small sample size and high attrition rate, limit the capacity of research teams to develop robust multivariate statistical models adjusted for confounding variables related to patient, interprofessional teams, and organizational characteristics [21, 23]. Interestingly, retrospective observational studies have been shown to be a relevant alternative to examine the association between ACNP and patient and organizational outcomes in postoperative cardiac surgery settings. Existing studies [27–30] have used retrospective patient cohorts to develop models adjusted for confounding variables. These studies [6, 27–29] have identified significant associations between interprofessional teams with ACNPs and lower mortality risk and decreased costs in postoperative cardiac surgery settings.

Thirdly, the majority of the RCTs focus on NP practice within primary care settings (e.g., in-home care, rehabilitation clinic) after the cardiac surgery. Less attention has been paid to ACNP practice in acute care settings; however, within these settings, ACNPs work in collaboration with cardiac surgeons, nursing teams, rehabilitation teams, social workers, and other members of interprofessional teams to ensure the patient's optimal recovery, and reduce adverse events and postoperative complications. For patients and families, hospitalization in acute care represents a crucial phase of postoperative recovery associated with several physical, psychosocial, and emotional stressors [30]. Additional studies are needed to substantiate the contribution of ACNPs in interprofessional teams within acute care settings, and identify the benefits of their practice for patients, families, interprofessional teams, and healthcare organizations. The aim of this project is to measure and examine the level of adherence to best-practice guidelines of interprofessional teams with and without ACNPs in an acute postoperative cardiac surgery unit in Québec, Canada.

## Study hypothesis and objectives

The hypothesis of this study is:

H1: It is hypothesized that interprofessional teams with ACNP are associated with a higher level of adherence to best-practice guidelines compared to interprofessional teams without ACNP, after controlling for the patient and interprofessional team characteristics.

The objectives of this study are as follows:

1. Develop and pilot-test an extraction tool to measure the level of adherence to best-practice guidelines of interprofessional teams within a postoperative cardiac surgery setting.

2. Describe the patient and interprofessional teams' characteristics, as well as the level of adherence to best-practice guidelines, of patients under the care of interprofessional teams with and without ACNPs.

3. Examine the association between interprofessional teams with ACNPs and the level of adherence to best-practice guidelines, compared to interprofessional teams without ACNPs, after adjusting for patient and interprofessional team characteristics in postoperative cardiac surgery.

## Methods and design

### Study design

A retrospective observational study [31, 32] will be conducted. De-identified data will be extracted from the University healthcare centre (UHC) data warehouse and the patient health records to assemble the retrospective cohort of patients, and conduct the data collection. Ethics

approval was obtained from the McGill University Health Centre Research Ethics Board on September 15, 2021 (IPSSA chirurgie cardiaque/2022-8094). The *Strengthening the Reporting of Observational Studies in Epidemiology (STROBE) Statement: Guidelines for Reporting Observational Studies* [32] will be used as a framework to conduct this project (S1 Appendix).

## Study setting

This study will be conducted at a UHC in Québec, Canada, where approximately 1,000 cardiac surgeries are performed annually in a 36-bed postoperative cardiac surgery unit.

After cardiac surgery, patients are admitted to the intensive care unit (ICU), and are under the care of ICU teams. At the time of data collection, the ICU teams in Québec, Canada did not include ACNPs. On average, patients are hospitalized for 24 hours in the ICU before being transferred to the postoperative cardiac surgery unit. Patients readmitted to the ICU are under the care of ICU teams.

Upon admission to the postoperative cardiac surgery unit, patients are assigned to the care of interprofessional teams with or without ACNPs. Team assignment depends on the current workload of each team, and the availability of the beds in the unit. Interprofessional teams with ACNPs are responsible for 16 of the unit's beds (44%) and interprofessional teams without ACNP are assigned 20 of the unit's beds (56%). Patients are followed by their respective teams from their admission to the cardiac surgery unit until hospital discharge or death. On average, the length of stay at the hospital after the surgery ranges from 9 to 11 days [33].

Interprofessional teams with and without ACNPs include the following: cardiac surgeons and physician residents, nursing teams, skin and wound care specialists, physiotherapists, nutritionists, respiratory therapists, social workers, and other medical specialists. Interprofessional teams with and without ACNPs ensure daily follow-up of patients, assessment of the patient's clinical and psychosocial condition, management of pharmacotherapy, clinical interventions, prescribing and monitoring of laboratory tests, lifestyle promotion, and the preparation of patients for hospital discharge.

## The practice of acute care nurse practitioners in the postoperative cardiac surgery unit

Since 2017, seven ACNPs practice in postoperative cardiac surgery, based on a monthly rotation system. Two ACNPs simultaneously practice within interprofessional teams and collaborate with other clinicians to ensure the daily follow-up of the patients after cardiac surgery. In addition to the activities described above, the ACNPs also teach coping strategies, lifestyle promotion, cardiac rehabilitation, preparation of the patients and families for the in-home recovery after discharge. The ACNPs support the preparation for hospital discharge (e.g., prescription of the patient medication after discharge, summary for the family physician, etc.) [34].

## Study population and sample

A retrospective cohort of patients will be assembled with patients hospitalized at the postoperative cardiac surgery unit between January 1, 2019, and January 31, 2020. This timeframe will avoid the recruitment of patients during the COVID-19 pandemic, a window of time during which the practice of ACNPs within the unit was inconsistent. Patients will be selected, based on three inclusion criteria: 1) patient admitted with a diagnosis of coronary artery bypass graft (CABG) and/or valve repair, 2) patient age is over 18 years old, and 3) patient has been hospitalized for at least 24 hours in the postoperative cardiac surgery unit. The minimum 24-hour criteria will allow the research team to examine the hospitalization of the patient in the

postoperative cardiac surgery unit, which is the setting where ACNPs practice in Québec [35]. Patients will be identified from the electronic databases of the UHC and the diagnostic codes of the Canadian Institute for Health Information [36] (S2 Appendix).

A sample size calculation was conducted in G*power [37]. This project will include 15 variables, including one independent variable, one dependent variable, and 13 confounding variables. The variables selected for this project are described in the next section. To perform one multivariate linear regression model and detect an effect size of $f^2$ = 0.15 with a power of 0.80 and a standard error of 5%, a minimum sample size of 183 patients was estimated.

A total of 300 patients will be selected, based on the sample size calculation and an overestimation of at least 20%, to consider the risk of error from the presence of missing data and incomplete patient health records [38, 39]. A systematic random selection of 150 patients under the care of interprofessional teams with ACNP will be conducted and matched with 150 patients under the care of interprofessional teams without ACNP. Three criteria [40–43] will be used simultaneously to match each pair (1:1): 1) age (i.e., a five-year gap will be tolerated) 2) sex, and 3) type of cardiac surgery (i.e., CABG, valve repair, or CABG/valve repair).

Patients will be followed from their admission to the ICU after the cardiac surgery, until the occurrence of the following events, whichever occurs first: 1) discharge from the hospital, 2) death, or 3) the cumulation of 14-days in hospital after the surgery. The 14-days' timeframe will allow the research team to capture the acute phase of the patient's hospitalization after the cardiac surgery [33]. The total length of stay for patients hospitalized longer than 14-days will be measured.

## Variables under study

A systematic review of RCTs [24] and an extensive search in the literature and international healthcare organizations (e.g., American Heart Association, Society of Thoracic Surgeons) was conducted to retrieve best-practice guidelines for interprofessional teams in postoperative cardiac surgery settings, and confounding variables related to patient and interprofessional teams' characteristics. A total of 12 best-practice guidelines and 13 confounding variables were identified and are presented in Fig 1. The operationalization of each variable is presented in S3 Appendix.

**Independent variable: Interprofessional teams with and without acute care nurse practitioners.** A dichotomous variable will be created to measure the inclusion of at least one ACNP within the interprofessional team. For patients under the care of an interprofessional team with ACNPs, an additional descriptive variable will measure the number of days where the ACNP was involved in the daily follow-up of the patient, on the entire hospitalization at the postoperative cardiac surgery unit.

**Dependant variable: The level of adherence to best-practice guidelines by interprofessional teams.** A total of 12 best-practice guidelines will be included in this project (Fig 1). All best-practice guidelines are divided into three categories: 1) pharmacotherapy (n = 4), 2) laboratory tests (n = 4), and 3) postoperative assessment (n = 4).

Four best-practice guidelines are included in the category of pharmacotherapy. These guidelines are the prescription and relevant monitoring of medications including: 1) anticoagulants, 2) beta blockers, 3) lipid-lowering agents, and 4) anti-platelets, during hospitalization and at hospital discharge of the patient. The adherence to these four guidelines is defined by the prescription (e.g., beta blocker) and relevant monitoring (e.g., assessment of the blood pressure) of those four medications on a daily basis during the patient's hospitalization and at discharge. For each best-practice guideline, two additional descriptive variables will be collected, including the type of medication prescribed and the presence of a contraindication, which precludes the possibility of achieving the guideline recommendation (e.g., allergy).

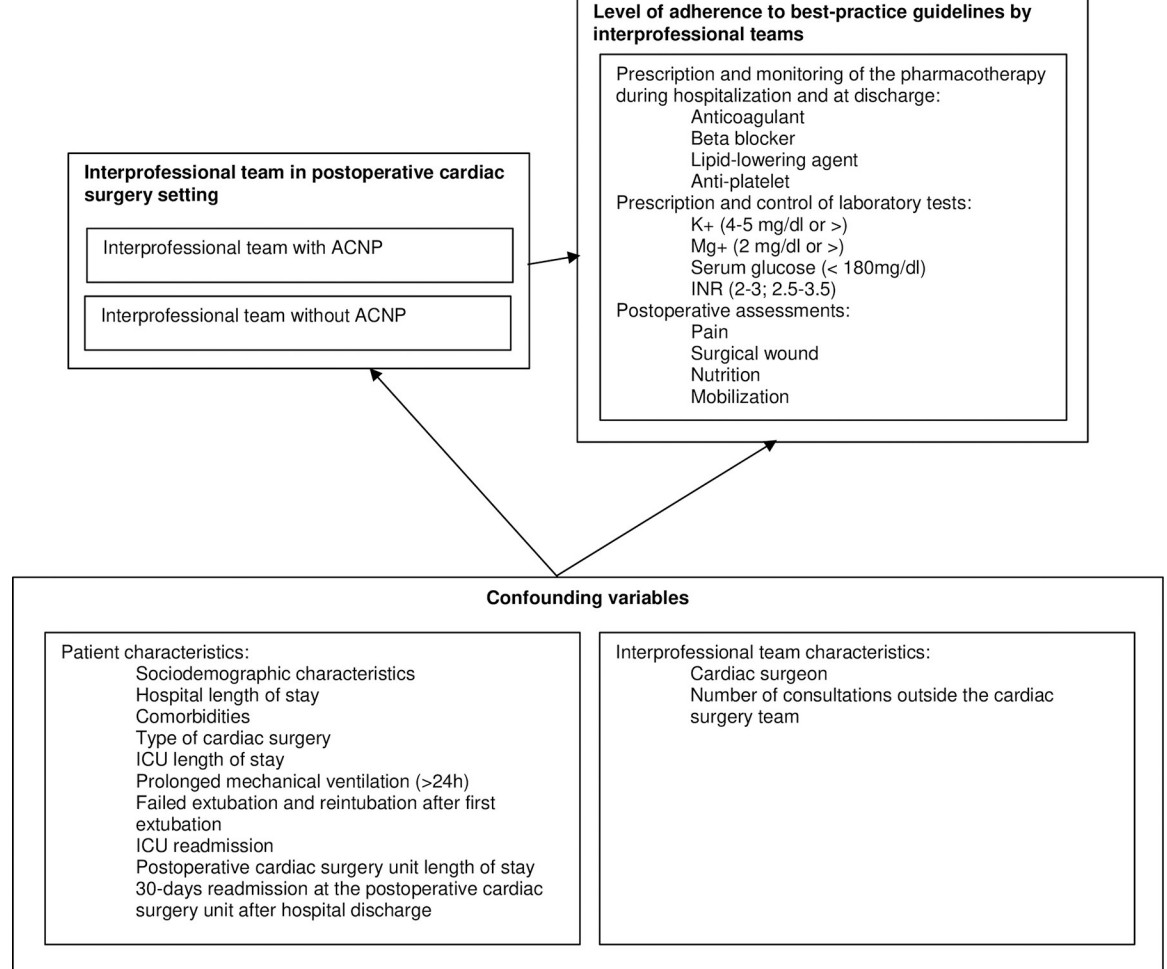

**Fig 1. The influencing variables and the level of adherence to best-practice guidelines by interprofessional teams with and without acute care nurse practitioners in cardiac surgery.**

Four best-practice guidelines on laboratory tests are included, recommending the prescription and monitoring of the following: 1) potassium (K+), 2) magnesium (Mg+), 3) serum glucose, and 4) international normalized ratio (INR) by interprofessional teams during the patient's hospitalization in the postoperative cardiac surgery unit. The achievement of each best-practice guideline recommendation will be confirmed if the laboratory result falls under the normal range supported by the current literature. The monitoring of laboratory tests also includes the interventions conducted by clinicians to ensure the laboratory test values remain within the normal range, such as routinely ordering laboratory tests, assessing laboratory tests to confirm normal values and detect any abnormal values, and performing pharmacological and clinical interventions to address abnormal laboratory test values. An additional variable called "no INR needed" will be added specifically for the best-practice guideline related to the prescription and management of INR, to take into consideration patient who don't need a strict INR follow-up (e.g., administration of direct acting oral anticoagulant (DAOC)). A pilot-test will be performed to assess the characteristics of the patients hospitalized at the post-operative cardiac surgery unit and the feasibility of the data collection. The following three additional descriptive variables will be measured for each guideline: 1) number of prescriptions

requested by interprofessional teams, 2) proportion of abnormal values below or above normal range, and 3) average of all values of laboratory tests.

Four best-practice guidelines are included in the category of the postoperative assessment performed by interprofessional teams. A daily assessment of the patient's level of pain, surgical wound(s), nutrition, and mobilization will be measured. For each best-practice guideline recommendation, the achievement of the guideline will be confirmed if at least one postoperative assessment per day is performed by the ACNP or other members of the interprofessional teams during the patient's hospitalization in the postoperative cardiac surgery unit. The postoperative assessments include the assessment of the clinical manifestations, signs, and symptoms experienced by the patient (e.g., patient's decrease mobility), as well as the interventions conducted by clinicians following these assessments. Interventions include pharmacological interventions (e.g., prescription of analgesic), non-pharmacological interventions (e.g., prescription of a coronary diet), and clinician consultations (e.g., consultation with the nutritionist). Two additional descriptive variables will be measured for each best-practice guideline, including: 1) the total number of postoperative assessments performed by interprofessional teams during the hospitalization, and 2) the type of associated interventions conducted by clinicians (e.g., non-pharmacological, consultation, etc.).

A composite score will be developed to measure the level of adherence to best-practice guidelines by interprofessional teams with and without ACNPs. A composite score per patient per day of hospitalization in postoperative cardiac surgery will be developed and based on three consecutive steps. Firstly, for each best-practice guideline, a score on a scale of two will be created and points will be attributed for: 1) the achievement of the best-practice guideline recommendation, and 2) the associated intervention performed by clinicians. Points will be given based on the information contained in the patient's health record and electronic databases. A missed intervention will be considered as a lack of adherence to best-practice guidelines, and no point will be given.

Secondly, an overall composite score will be calculated from the sum of all the scores of all best-practice guidelines (n = 12), ranging from zero to 24. The overall composite score will be transformed into a value in percentage ranging from 0 to 100% (i.e., composite score/24 * 100), such that 100% represents a complete adherence (24/24). An increase of the score will be associated with an increase of the level of adherence to best-practice guidelines by interprofessional teams. An overall score will be calculated per patient per each day hospitalized in the postoperative cardiac surgery unit.

Finally, an average composite score will be calculated from the average of all overall scores during the patient's hospitalization at the postoperative cardiac surgery unit. An average composite score will be calculated for all patients included in the retrospective cohort.

**Variables influencing the effect of acute care nurse practitioner practice related to patient characteristics.**   Eleven influencing variables related to patient characteristics will be measured. For each patient, sociodemographic characteristics (e.g., sex, age) will be measured. The total length of stay at the hospital will be measured, which includes the stay in the ICU and postoperative cardiac surgery unit. The type of cardiac surgery will be collected from a categorical variable with three categories: 1) CABG, 2) valve repair, and 3) CABG/valve repairs. Comorbidities will be measured and operationalized from the Charlson Comorbidity Index (CCI) [44, 45]. A total score of 24 points will be calculated for each patient based on the 17 clinical conditions included in the CCI (S4 Appendix). The length of stay in the postoperative cardiac surgery unit will be collected for each patient. The hospital 30-days readmission at the postoperative cardiac surgery unit after discharge will be measured for each patient from a dichotomous variable.

Four influencing variables related to the patient characteristics in the ICU setting will be measured. The total number of hours hospitalized in the ICU will be measured from the time

(in hours) of admission to the ICU after the surgery and the time (in hours) of discharge of the patient. The postoperative prolonged mechanical ventilation will be measured from a dichotomous variable. A prolonged mechanical ventilation will be confirmed if the time under mechanical ventilation exceeds 24 hours. The failed extubation of the patient and reintubation after first extubation will be measured with a dichotomous variable. The number of episodes of ICU readmission after initial discharge will be measured for each patient.

**Variables influencing the effect of acute care nurse practitioner practice related to interprofessional teams' characteristics.**   Two influencing variables related to interprofessional teams' characteristics will be measured. First, the cardiac surgeon who performed the surgery and ensured the follow-up of the patient's recovery will be noted. A confidential code will be created for each surgeon practising in the UHC. Second, a discrete variable will be created to measure the number of consultations conducted by clinicians from different professional groups (e.g., nutrition, rehabilitation team) during the patient's hospitalization in the postoperative cardiac surgery unit. A categorical descriptive variable will be created to measure each clinician's professional group, including the following: 1) rehabilitation team (e.g., physical therapist, occupational therapist), 2) respiratory therapist, 3) social worker, 4) skin and wound care therapist, 5) nutritionist, and 6) speciality consultation (e.g., internal medicine, nephrology).

## Data collection

An extraction tool will be developed and pilot-tested, based on the best-practice guidelines and the confounding variables related to the patient and interprofessional team characteristics (Fig 1). One extraction will be completed per patient per day during the hospitalization in the postoperative cardiac surgery unit, by the first author or a research assistant. The development of the extraction tool followed the Consensus-based Standards for the selection of health measurement instrument (COSMIN) guidelines [46] and will include three stages: the content validation; the criterion validation; and the pilot-test.

**Data collection for the content validation.**   Content validation will be performed to assess the clinical relevance and the representativeness of the practice of interprofessional teams in post-operative cardiac surgery setting of the items of the extraction tool. Content validation will follow two steps [46]. First, an expert committee will be recruited from clinicians, managers, and researchers who have expertise in postoperative cardiac surgery. The identification and recruitment of experts will be conducted by our research team and will include experts in Canada. A total of 10 experts will be recruited, including at least the following five: 1) an ACNP who worked a minimum of one year within a postoperative cardiac surgery unit outside of our associated UHC, 2) a cardiac surgeon or fellow in cardiac surgery who worked at least two years in an acute care centre outside of our associated UHC, 3) a nurse manager who has led for at least one year, a postoperative cardiac surgery unit, within our associated UHC, 4) a healthcare professional who worked in the data warehouse of our associated UHC for at least one year, and 5) a researcher in nursing or healthcare sciences with an expertise in measurement instruments and/or cardiac surgery.

Second, an electronic survey [47] will be created and sent to each expert of the committee. The survey will be composed of all items of the extraction tool (best-practice guidelines and confounding variables). For each item, experts will assess its relevance using a 5-points Likert scale [48, 49]. Additional spaces for qualitative comments and suggestions will be included in the survey. Two rounds of revision are planned. After the first round, our team will update the extraction tool and revise any unclear items based on the experts' recommendations, and an updated version will be resubmitted to the expert committee.

**Data collection for the criterion validation.** The criterion validation of the extraction tool will follow two consecutive steps. Firstly, the best-practice guidelines of The Society of Thoracic Surgeons (STS) have been selected as the gold standard of comparison for this study. For many years, the STS has been a leading healthcare organization in the development and validation of best-practice guidelines and performance measures in the care of patients in cardiac surgery settings [10]. In 2007, the STS developed and validated 21 performance measures for the delivery of high quality of care for patients who underwent CABG, valve repair, and CABG/valve repairs [50, 51]. For this project, five performance measures were selected, including: 1) prolonged mechanical ventilation higher than 24 hours in the ICU; 2) a 30-day readmission rate at the postoperative cardiac surgery unit after hospital discharge; and prescription of the 3) anti-platelet, 4) beta blocker, 5) and lipid-lowering agent, during the hospitalization at the postoperative cardiac surgery unit and at discharge. Each performance measure is operationalized as the proportion of patients who achieved the outcome (e.g., proportion of patients who were under prolonged mechanical ventilation) over the total sample size [46, 47].

Secondly, 30 patient health records will be reviewed independently by two reviewers on two occasions, including the first author and a research assistant. For the first round of revision, the patient's health record will be reviewed with the extraction tool. For the second round of revision, the records will be reviewed with the five performance measures from the STS.

**Data collection for the pilot-test.** A pilot-test will be conducted to assess the feasibility of using the tool, standardize data collection among reviewers, examine the quality and accessibility of the retrospective data, and update the tool if needed. Four iterative steps will be included in the pilot-test [52–55]. First, two reviewers will conduct the pilot-test and data collection. Training sessions and documentation will be given to the reviewers.

Secondly, 30 patient health records from hospitalized patients in the postoperative cardiac surgery unit between January 1, 2019, and January 31, 2020, will be randomly selected. These patients will not be included in the retrospective cohort. Data will be collected by the two reviewers with the extraction tool. Inter-rater reliability will be assessed and a Cohen kappa higher than 0.60 will be targeted [55].

Thirdly, a triangulation of the available data will be performed to assess the quality and accessibility of the data. Multiple data sources will be screened between the electronic databases and patient health records. Items from the extraction tool with a high quantity of missing data or low quality of available data will be modified or removed.

Finally, frequent meetings among all research team members will be conducted to discuss divergences, reinforce convergences, and establish a consensus. The extraction tool will be updated after the pilot-test if needed [52].

A calibration session among the reviewers will be conducted during the data collection from the main retrospective cohort. A total of 30 patient health records will be reviewed by two independent reviewers [49, 50]. Inter-rater reliability will be assessed and a Cohen Kappa value higher than 0.60 will be targeted [55].

## Data analysis

Statistical analysis will be conducted to meet objectives one, two and three. The unit of observation will be the patient-level and the unit of analysis will be the interprofessional team with or without ACNP-level.

**Objective 1.** For the content validation [56, 57], a Fleiss Kappa will be calculated for each round of revision and a value between 0.60 and 0.80 will be targeted [48, 49]. The content validity index (CVI) of each individual item of the extraction tool will be calculated from the proportion of experts who rated a score of 4 or 5 on the Likert scale [58]. A CVI of 0.80 will be

targeted for each item and items with a value below 0.80 will be removed or modified. Then, the proportion of items with a CVI value above 0.80 will be calculated to determine the overall CVI of the extraction tool. A value higher than 0.80 for the overall CVI of the extraction tool will be targeted [46]. Qualitative comments and recommendations by experts will be analyzed by content analysis [59].

For the criterion validation, Spearman correlation will be calculated to assess the correlation between the measures of the gold standard (e.g., the performance measures of the STS), and the measures of the extraction tool. Values of Spearman correlation higher than 0.70 will be targeted [48, 49].

**Objective 2.**　Descriptive analysis will be performed to assess the level of adherence by interprofessional teams with and without ACNPs, as well as the confounding variables for patients under the care of both teams [60]. A graphic illustration will be created to visualize the tendency and distribution of the level of adherence to best-practice guidelines by interprofessional teams with and without ACNPs. An analysis of missing data will be conducted to examine the quality and distribution of missing data within patient health records and electronic databases. Team meetings and consultations with clinicians and statisticians will be held to discuss and identify potential causes of missing data, as well as to conduct a statistical analysis to manage them [61]. Bivariate analysis will be conducted, and the statistical significance will be based on a $p$-value of 0.05 [60]. The Bonferroni correction will be applied for the examination of each confounding variable related to the patient and interprofessional teams' characteristics [62].

**Objective 3.**　A multivariate linear regression model will be developed to examine the association between interprofessional teams with and without ACNPs and the level of adherence to best-practice guidelines [63–65]. An examination of the distribution of the data will be performed. In the case of an abnormal distribution of the data, a transformation will be conducted to adjust the statistical model based on the observed data. The independent variables will include the interprofessional teams with and without ACNPs, as well confounding variables. A correlation matrix will be developed, and tolerance value lower than 0.2 and a variance inflation factor (VIF) value higher than 5 will be used to identify multicollinearity between independent variables [63–65]. Independent variables with multicollinearity will be removed one at the time and a new regression will be calculated. The dependent variable of the regression model will be the average composite score (%) of the level of adherence to best-practice guidelines. Statistical significance will be based on a $p$-value of 0.05. Sensitivity analysis will be conducted based on a selection of confounding variables related to the characteristics of the patient (e.g., length of stay, ICU readmission). The selection of these confounding variables will be based on the observed data and descriptive analysis. The residual analysis and the analysis of the $R^2$ will be conducted to assess the goodness-of-fit of the multivariate linear regression model.

## Interpretation of the findings

Consultations with clinicians and managers practicing in postoperative cardiac surgery settings in Québec and Canada will be conducted to gain an in-depth understanding of the findings. These consultations will allow the research team to explore potential underlying factors and mechanisms which could influence the level of adherence to best-practice guidelines by interprofessional teams with and without ACNPs. Field notes will be taken during these consultations and analyzed using content analysis [45].

## Ethical considerations

Ethical approval was obtained prior to the beginning of the study. No identifying information will be collected during the content validation with the expert committee and the data

collection within electronic databases and patient health records. A confidential pairing system will be created to match the patient identification with a random number [66]. The database of the project will be kept in a secure server, protected by a confidential numeric code known only by the research team. During the dissemination of the findings, data will be shared in an aggregated form and the identity of the UHC, as well as the identity of the patients and clinicians, will remain confidential.

## Discussion

ACNPs hold an important place within interprofessional teams caring for patients and families following cardiac surgery. For patients and families, ACNPs contribute to the management of pharmacotherapy, clinical and psychosocial conditions, as well as lifestyle promotion and teaching of coping strategies. For interprofessional teams, ACNPs enhance the collaboration and communication among team members and reinforce continuity of care. Until now studies examined the efficacy of ACNPs on patient and organization outcomes within postoperative cardiac surgery settings; however, less attention has been paid to underlying factors contributing to the effect of ACNPs on these outcomes.

The findings of this study will further our understanding of the contributions of ACNPs in interprofessional teams practising in postoperative cardiac surgery settings in Québec, Canada. This project will generate empirical data to support and document the practice of ACNPs within interprofessional teams in three important ways. Firstly, an extraction tool focusing on the level of adherence to best-practice guidelines by interprofessional teams in postoperative cardiac surgery will be available for the field of cardiac surgery research and the scientific community more broadly. Secondly, for patients under the care of interprofessional teams with and without ACNPs, the findings will provide a description of the patient and interprofessional team characteristics as well as the adherence to best-practice guidelines. Finally, this study will examine the effect of ACNPs on the adherence to best-practice guidelines in settings where ACNPs are incorporated within inter-professional teams, and provide empirical data for those settings looking to introduce such roles in a postoperative cardiac surgery setting.

## Supporting information

**S1 Appendix. STROBE statement—checklist of items that should be included in reports of observational studies.**
(DOCX)

**S2 Appendix. Diagnosis codes from the Canadian classification of health information.**
(DOCX)

**S3 Appendix. Variables under study.**
(DOCX)

**S4 Appendix. Charlson comorbidity index.**
(DOCX)

## Acknowledgments

The authors would like to acknowledge the comments received from Christine Maheu, Andraea Van Hulst, and Alain Biron.

## Author Contributions

**Conceptualization:** Li-Anne Audet, Mélanie Lavoie-Tremblay, Éric Tchouaket, Kelley Kilpatrick.

**Methodology:** Li-Anne Audet, Mélanie Lavoie-Tremblay, Éric Tchouaket, Kelley Kilpatrick.

**Supervision:** Mélanie Lavoie-Tremblay, Éric Tchouaket, Kelley Kilpatrick.

**Writing – original draft:** Li-Anne Audet.

**Writing – review & editing:** Mélanie Lavoie-Tremblay, Éric Tchouaket, Kelley Kilpatrick.

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
