## [Decision Letter · Decision Letter 0]

11 Jul 2022

PONE-D-21-40085The level of adherence to best-practice guidelines by interprofessional teams with and without acute care nurse practitioners in cardiac surgery: a study protocolPLOS ONE

Dear Dr. Audet,

Thank you for submitting your manuscript to PLOS ONE. After careful consideration, we feel that it has merit but does not fully meet PLOS ONE’s publication criteria as it currently stands. Therefore, we invite you to submit a revised version of the manuscript that addresses the points raised during the review process.

Please note that we have only been able to secure a single reviewer to assess your manuscript. We are issuing a decision on your manuscript at this point to prevent further delays in the evaluation of your manuscript. Please be aware that the editor who handles your revised manuscript might find it necessary to invite additional reviewers to assess this work once the revised manuscript is submitted. However, we will aim to proceed on the basis of this single review if possible.

The reviewer's comments may be seen below. They are very positive regarding the topic and overall reporting of the mansucript, and have also provided some minor suggests for the formatting and presentation of information within the paper. Furthermore they have recommended copy editing for English language presentation.

Could you please revise the manuscript to carefully address the concerns raised?

We look forward to receiving your revised manuscript.

Kind regards,

Lucinda Shen, MSc

Staff Editor

PLOS ONE

Journal Requirements:

2, Your ethics statement should only appear in the Methods section of your manuscript. If your ethics statement is written in any section besides the Methods, please delete it from any other section.

4 Please review your reference list to ensure that it is complete and correct. If you have cited papers that have been retracted, please include the rationale for doing so in the manuscript text, or remove these references and replace them with relevant current references. Any changes to the reference list should be mentioned in the rebuttal letter that accompanies your revised manuscript. If you need to cite a retracted article, indicate the article’s retracted status in the References list and also include a citation and full reference for the retraction notice.

Reviewers' comments:

Reviewer's Responses to Questions

**Comments to the Author**

1. Does the manuscript provide a valid rationale for the proposed study, with clearly identified and justified research questions?

Reviewer #1: Yes

2. Is the protocol technically sound and planned in a manner that will lead to a meaningful outcome and allow testing the stated hypotheses?

Reviewer #1: Yes

3. Is the methodology feasible and described in sufficient detail to allow the work to be replicable?

Reviewer #1: Yes

4. Have the authors described where all data underlying the findings will be made available when the study is complete?

Reviewer #1: Yes

5. Is the manuscript presented in an intelligible fashion and written in standard English?

Reviewer #1: Yes

6. Review Comments to the Author

You may also provide optional suggestions and comments to authors that they might find helpful in planning their study.

Reviewer #1: Thank you for your submission to PLOS ONE. The comments provided in this review are done so in the spirit of strengthening the quality of your manuscript and the scholarly discourse that this paper will provide readers of this journal.

I encourage you to revise your opening paragraph to acknowledge the fact that cardiac surgery is a service provided primarily in developed nations. Whilst there are numerous procedures performed annually globally, there are swathes of people in undeveloped nations that do not have equitable access to prevention, diagnosis or surgical intervention for cardiovascular diseases. In addition, specific discourse addressing the fact that nurses are almost always a component of multi-disciplinary or inter-professional teams, but very few nurses are provided opportunities for independent advanced practice. It is pleasing to see that in the setting for this study the role of the acute care nurse practitioner is valued, as evidenced by the number of these nurses in the postoperative cardiac surgical setting for the study. This needs to be contrasted to other national and international settings providing similar services as a mechanism of highlighting the fact that there has been a commitment to innovative thinking by executive leaders in the organisation in question.

In the methods section of your paper present your hypothesis before specific study objectives as this relates to your primary study aim.

In your concluding remarks when you justify the potential benefits associated with this study, please ensure you reiterate all of the outcomes associated with study objectives.

Throughout the paper there are very minor grammatical and syntax errors that should be amended.

The in-text citation numbers are out of sync and need to be amended.

7. PLOS authors have the option to publish the peer review history of their article (what does this mean?). If published, this will include your full peer review and any attached files.

Reviewer #1: **Yes: **Dr Rochelle Wynne, Clinical Nurse Consultant, Royal Melbourne Hospital, Parkville, Australia.

Honorary Professor of Nursing, School of Nursing, University of Wollongong.

Honorary A/Professor of Nursing, School of Nursing & Midwifery, Deakin University.

---

## [Author Response · Author response to Decision Letter 0]

13 Sep 2022

September 13th, 2022

To the Editor of Plos One, 

We thank the reviewer and the editor for their thoughtful comments and have summarized our modifications to the manuscript addressing their questions and suggestions in the table attached with our submission. 

Please do not hesitate to contact us if you require any additional information or have any further questions. 

We will look forward to hearing from you. 

Sincerely, 

Li-Anne Audet

---

## [Decision Letter · Decision Letter 1]

14 Nov 2022

PONE-D-21-40085R1The level of adherence to best-practice guidelines by interprofessional teams with and without acute care nurse practitioners in cardiac surgery: a study protocolPLOS ONE

Dear Dr. Audet,

Thank you for submitting your manuscript to PLOS ONE. After careful consideration, we feel that it has merit but does not fully meet PLOS ONE’s publication criteria as it currently stands. Therefore, we invite you to submit a revised version of the manuscript that addresses the points raised during the review process.

Please revise.

We look forward to receiving your revised manuscript.

Kind regards,

Academic Editor

PLOS ONE

Journal Requirements:

Reviewers' comments:

Reviewer's Responses to Questions

**Comments to the Author**

1. Does the manuscript provide a valid rationale for the proposed study, with clearly identified and justified research questions?

Reviewer #1: Yes

Reviewer #2: Yes

2. Is the protocol technically sound and planned in a manner that will lead to a meaningful outcome and allow testing the stated hypotheses?

Reviewer #1: Partly

Reviewer #2: Yes

3. Is the methodology feasible and described in sufficient detail to allow the work to be replicable?

Reviewer #1: Yes

Reviewer #2: Yes

4. Have the authors described where all data underlying the findings will be made available when the study is complete?

Reviewer #1: No

Reviewer #2: Yes

5. Is the manuscript presented in an intelligible fashion and written in standard English?

Reviewer #1: Yes

Reviewer #2: Yes

6. Review Comments to the Author

You may also provide optional suggestions and comments to authors that they might find helpful in planning their study.

Reviewer #1: Please find attached a PDF file with annotations and comments for your consideration.

There are some methodological inconsistencies in this manuscript that need to be further clarified.

The Data Analysis section is largely repeated in the data collection section and the relevant text for these two sections should be distinct. In part the repetition that is evident may be due to study design considerations.

I do believe this paper is worthy of publication so ask that you consider the feedback that will strengthen the work and also benefit next steps in terms of conducting the study.

In your response I ask that you provide a 3 column table noting the feedback/questions that require a response in the first column, your comments/corrections relevant to that point in the second column and then the page and line number of the change in the next revision. This approach enables a much more efficient review.

I look forward to reading your next revision.

Reviewer #2: The Authors submitted a revised version of their previously submitted Study Protocol paper. The manuscript and the protocollo significantly improved after the revision. I have no further comments or edits. The authors should be commende for their efforts..

7. PLOS authors have the option to publish the peer review history of their article (what does this mean?). If published, this will include your full peer review and any attached files.

Reviewer #1: No

Reviewer #2: **Yes: **Francesco Bianco

---

## [Author Response · Author response to Decision Letter 1]

19 Dec 2022

December 19th, 2022

To the Editor of Plos One, 

We thank the reviewers and the editor for their thoughtful comments and have summarized our modifications to the manuscript addressing their questions and suggestions in the table that follow. 

Please do not hesitate to contact us if you require any additional information or have any further questions. 

We will look forward to hearing from you. 

Sincerely, 

Li-Anne Audet

---

## [Decision Letter · Decision Letter 2]

11 Jan 2023

PONE-D-21-40085R2The level of adherence to best-practice guidelines by interprofessional teams with and without acute care nurse practitioners in cardiac surgery: a study protocolPLOS ONE

Dear Dr. Audet,

Thank you for submitting your manuscript to PLOS ONE. After careful consideration, we feel that it has merit but does not fully meet PLOS ONE’s publication criteria as it currently stands. Therefore, we invite you to submit a revised version of the manuscript that addresses the points raised during the review process.

Please revise.

We look forward to receiving your revised manuscript.

Kind regards,

Academic Editor

PLOS ONE

Reviewers' comments:

Reviewer's Responses to Questions

**Comments to the Author**

1. Does the manuscript provide a valid rationale for the proposed study, with clearly identified and justified research questions?

Reviewer #2: Yes

Reviewer #3: Yes

Reviewer #4: Yes

2. Is the protocol technically sound and planned in a manner that will lead to a meaningful outcome and allow testing the stated hypotheses?

Reviewer #2: Yes

Reviewer #3: Partly

Reviewer #4: Yes

3. Is the methodology feasible and described in sufficient detail to allow the work to be replicable?

Reviewer #2: Yes

Reviewer #3: Yes

Reviewer #4: Yes

4. Have the authors described where all data underlying the findings will be made available when the study is complete?

Reviewer #2: Yes

Reviewer #3: Yes

Reviewer #4: No

5. Is the manuscript presented in an intelligible fashion and written in standard English?

Reviewer #2: Yes

Reviewer #3: Yes

Reviewer #4: Yes

6. Review Comments to the Author

You may also provide optional suggestions and comments to authors that they might find helpful in planning their study.

Reviewer #2: The Authors submitted a Study Protocol paper centered on the level of adherence to best-practice guidelines by interprofessional teams with and without acute care nurse practitioners in cardiac surgery. The study would be a retrospective observational study enrolling 300 patients hospitalized between January 1, 2019, and January 31, 2020, in a postoperative cardiac surgery unit in Québec, Canada.

To the best of my knowledge, the protocol has its merits. It would represent the first study to examine the level of adherence to best-practice guidelines by interprofessional teams with and without ACNP in a postoperative cardiac surgery setting.

The manuscript is well written, even if more attention should be given to English grammar and structure. Some typos should be addressed too (i.e., line 340 double parenthesis after DOAC). The best-practice guidelines the Authors refer to should be explained in the text in extenso and not only in the references. Finally, it should be given the measure of the level of adherence or better highlighted since it needs to be clarified.

Reviewer #3: Dear Sir/MAdam,

Greetings!

A well designed protocol. But, the title is too long and confusing.

Best Wishes,

Reviewer #4: First of all, it was a pleasure reviewing your manuscript. Second, I found the topic to be very interesting. The focus on the roles of ACNPs in medical teams is important and shows the value of these individuals in providing high quality care with better health outcomes without undermining the roles of other healthcare professionals.

Third, I found this protocol to be very detailed and included all necessary information and I am excited to see how this study will turn out.

Finally, I only have some comments regarding some the content of the manuscript. In line 137-140, I found it difficult to understand the meaning and purpose of the sentence. I would recommend rewriting this part to make it clear. In line 179, there are some typing errors that need adjustment. In line 525-526, you mentioned the possibility of conducting data transformation in the case of abnormal distribution of the data. I would recommend that you carefully consider this approach as it is a sensitive matter and requires much caution when interpreted.

Thank you and please accept my best regards,

7. PLOS authors have the option to publish the peer review history of their article (what does this mean?). If published, this will include your full peer review and any attached files.

Reviewer #2: **Yes: **Francesco Bianco

Reviewer #3: No

Reviewer #4: No

---

## [Author Response · Author response to Decision Letter 2]

31 Jan 2023

January 31st, 2023

To the Editor of Plos One, 

We thank the reviewers and the editor for their thoughtful comments and have summarized our modifications to the manuscript addressing their questions and suggestions in the table that follows. 

Please do not hesitate to contact us if you require any additional information or have any further questions. 

We will look forward to hearing from you. 

Sincerely, 

Li-Anne Audet

---

## [Decision Letter · Decision Letter 3]

16 Feb 2023

The Level of Adherence to Best-Practice Guidelines by Interprofessional Teams With and Without Acute Care Nurse Practitioners in Cardiac Surgery: A Study Protocol

PONE-D-21-40085R3

Dear Dr. Audet,

We’re pleased to inform you that your manuscript has been judged scientifically suitable for publication and will be formally accepted for publication once it meets all outstanding technical requirements.

Kind regards,

Academic Editor

PLOS ONE

Additional Editor Comments (optional):

Reviewers' comments:

Reviewer's Responses to Questions

**Comments to the Author**

1. Does the manuscript provide a valid rationale for the proposed study, with clearly identified and justified research questions?

Reviewer #2: Yes

Reviewer #3: Yes

2. Is the protocol technically sound and planned in a manner that will lead to a meaningful outcome and allow testing the stated hypotheses?

Reviewer #2: Yes

Reviewer #3: Yes

3. Is the methodology feasible and described in sufficient detail to allow the work to be replicable?

Reviewer #2: Yes

Reviewer #3: Yes

4. Have the authors described where all data underlying the findings will be made available when the study is complete?

Reviewer #2: Yes

Reviewer #3: No

5. Is the manuscript presented in an intelligible fashion and written in standard English?

Reviewer #2: Yes

Reviewer #3: Yes

6. Review Comments to the Author

You may also provide optional suggestions and comments to authors that they might find helpful in planning their study.

Reviewer #2: The Manuscript and the protocol significantly improved after the revisions; I have no further comments or edits.

Reviewer #3: Much better after revision!

Much better after revision!

Much better after revision!

Much better after revision!

7. PLOS authors have the option to publish the peer review history of their article (what does this mean?). If published, this will include your full peer review and any attached files.

Reviewer #2: **Yes: **Francesco Bianco

Reviewer #3: No

---

## [Editor Report · Acceptance letter]

20 Feb 2023

PONE-D-21-40085R3 

The Level of Adherence to Best-Practice Guidelines by Interprofessional Teams With and Without Acute Care Nurse Practitioners in Cardiac Surgery: A Study Protocol 

Dear Dr. Audet:

I'm pleased to inform you that your manuscript has been deemed suitable for publication in PLOS ONE. Congratulations! Your manuscript is now with our production department. 

Kind regards, 

on behalf of

Dr. Robert Jeenchen Chen 

Academic Editor

PLOS ONE